# The Impact of Air Pollution on the Lung–Gut–Liver Axis: Oxidative Stress and Its Role in Liver Disease

**DOI:** 10.3390/antiox14101148

**Published:** 2025-09-23

**Authors:** Jacopo Iaccarino, Irene Mignini, Rossella Maresca, Gabriele Giansanti, Giorgio Esposto, Raffaele Borriello, Linda Galasso, Maria Elena Ainora, Antonio Gasbarrini, Maria Assunta Zocco

**Affiliations:** CEMAD Digestive Diseases Center, Fondazione Policlinico Universitario “A. Gemelli” IRCCS, Università Cattolica del Sacro Cuore, Largo A. Gemelli 8, 00168 Rome, Italy; jacopo.iaccarino01@icatt.it (J.I.); irene.mignini@guest.policlinicogemelli.it (I.M.); rossella.maresca12@gmail.com (R.M.); gabrielegiansanti.gg@gmail.com (G.G.); giorgio.esposto@guest.policlinicogemelli.it (G.E.); raffaeleborr@gmail.com (R.B.); linda.galasso0817@gmail.com (L.G.); mariaelena.ainora@policlinicogemelli.it (M.E.A.); antonio.gasbarrini@unicatt.it (A.G.)

**Keywords:** lung–gut–liver axis, air pollution, oxidative stress, liver disease

## Abstract

The expression “lung–gut–liver axis” refers to the interconnected processes occurring in the lungs, gastrointestinal tract, and liver, particularly in relation to immune function, microbial regulation, and metabolic responses. Over the past decade, growing concern has emerged regarding the detrimental impact of air pollution on liver disease. Air pollutants, including particulate matter (PM) and chemical gases such as nitrogen oxides (NOx), can influence the microbiome in the lungs and gut by generating reactive oxygen species (ROS), which induce oxidative stress and local inflammation. This redox imbalance leads to the production of altered secondary microbial metabolites, potentially disrupting both the alveolar–capillary and gut barriers. Under these conditions, microbes and their metabolites can translocate to the liver, triggering inflammation and contributing to liver diseases, particularly metabolic dysfunction-associated steatotic liver disease (MASLD), cirrhosis, and hepatocellular carcinoma (HCC). This manuscript aims to review recent findings on the impact of air pollution on liver disease pathogenesis, exploring the molecular, genetic, and microbiome-related mechanisms underlying lung–gut–liver interactions, providing insights into potential strategies to prevent or mitigate liver disease progression.

## 1. Introduction

Air pollution comprises a range of deleterious agents, including particulate matter (PM), nitrogen-based compounds such as nitrogen oxides (NOx) and nitrogen dioxide (NO_2_), ozone (O_3_), sulfur dioxide (SO_2_), and toxic metals. These pollutants can disrupt physiological homeostasis and contribute to conditions such as gut microbiota imbalance and oxidative stress [1]. PM is a heterogeneous mixture of chemicals, mineral particles, microorganisms, and organic materials. Based on its diameter, PM is classified into PM_10_ (particles smaller than 10 µm), PM_2.5_ (particles smaller than 2.5 µm, often referred to as “fine” particles), PM_2.5–10_ (particles whose diameter is between 2.5 and 10 µm) and ultrafine particles (UFPM, diameter < 100 nm) [1]. Smaller particles, such as PM_2.5_, may be transported from the lungs to the gastrointestinal tract via mucociliary clearance, where they can directly affect the gut microbiome [1,2], thus representing a link between respiratory and gastrointestinal systems. Moreover, gut microbiota imbalance can lead to increased oxidative stress in the liver, which may result in inflammation and fibrotic alterations [3]. Oxidative stress and direct contact with PM have also been linked to the onset of steatotic liver and its advancement to cirrhosis [4,5].

The sophisticated network of interactions among lung, gut and liver is named “lung–gut–liver axis” and involves immune signaling systems, metabolic pathways, and microbial products, which are essential for systemic homeostasis and immune response [6]. Altered gut microbiota composition and gut barrier have been associated with several liver and lung disorders [7,8]. Hence, the gut–liver–lung axis becomes critically relevant when considering the systemic effects of environmental exposures, especially air pollution. Inhaled pollutants not only directly damage the lungs but also compromise intestinal and hepatic barrier integrity. These systemic disturbances can amplify oxidative stress, promote chronic inflammation, and accelerate the progression of both liver and lung diseases. Previous reviews on the topic have primarily summarized the epidemiological and experimental evidence linking air pollution to liver, mainly metabolic-associated liver dysfunction [5,9]. In contrast, this work specifically delineates the role of oxidative stress, dysbiosis, and epithelial barrier dysfunction as key interconnected mechanisms mediating the hepatic consequences of chronic air pollution exposure, providing insights into some potential strategies to mitigate liver disease progression. The following sections aim to deepen the understanding of the complex lung–gut–liver axis, explore the specific effects of air pollutants on the gut and lung microbiota, examine the role of oxidative stress in pollution-induced liver injury, and review the growing body of evidence linking air pollution to liver disorders ranging from steatosis to hepatocellular carcinoma (HCC). Finally, we will analyze the potential benefits of microbiota modulation and antioxidant supplementation, paving the way to future treatments or prevention strategies against pollution-mediated liver injury.

## 2. Lung–Gut–Liver Axis: A Tri-Organ Crosstalk in Immune and Metabolic Regulation

The concept of lung–gut–liver axis has emerged since 2015, following the evidence that high dietary fiber intake was inversely correlated with chronic obstructive pulmonary disease (COPD) risk. To explain such correlation, it has been hypothesized that dietary fibers may have immunomodulatory properties and the capacity to influence liver macrophage activation, ultimately reducing systemic and pulmonary inflammation. Upon further examination, dietary fiber proved to exert positive effects on pulmonary function by promoting the proliferation of beneficial gut microbiota, which in turn enhances the production of short-chain fatty acids (SCFAs), metabolites with significant physiological and immunomodulatory functions [6]. Through the portal vein circulation, SCFAs inhibit 3-hydroxy-3-methylglutaryl-coenzyme A reductase, a critical metabolic enzyme implicated in the regulation of hepatic immune responses [10], thus attenuating innate immune system and decreasing blood levels of pro-inflammatory cytokines such as interleukin-6 (IL-6). Furthermore, SCFAs are also involved in lung homeostasis and carcinogenesis by inhibiting histone deacetylase which regulates immune T cells, and SCAFs levels are lower in COPD patients [8]. Indeed, a large prospective cohort study in the United Kingdom including ~120,000 participants found that adherence to a healthy dietary pattern was associated with a 33% reduction in COPD incidence (Hazard Ratio—HR = 0.67) [11]. Additionally, epidemiological data support the notion that high intake of dietary fiber, catechins, and flavonoids may protect against respiratory morbidity. The MORGEN study revealed that higher consumption of these compounds correlated with improved lung function (Forced Expiratory Volume in 1 s—FEV_1_) and a lower prevalence of COPD-related symptoms, such as chronic cough and dyspnea [12].

The gut microbiota is closely connected to the liver through the portal vein circulation and the enterohepatic bile acid pathway. Dysbiosis and impaired gut barrier function can facilitate the translocation of microbial products into the bloodstream, triggering hepatic inflammation and immune activation [7,13]. A review on this subject by Ha S. et al. underlines how gut microbiota and its metabolites may induce host epigenetic alterations with a detrimental impact on liver disease, notably metabolic dysfunction-associated steatotic liver disease (MASLD) and HCC. Gut microbiota affects DNA methylation through hypermethylation or hypomethylation, which regulate gene expression by silencing or activating them, respectively. In MASLD, gut dysbiosis is associated with DNA methylation of genes involved in lipogenesis [14]. The liver, in turn, plays a central role in regulating systemic immune responses by acting as a reservoir of tissue-resident macrophages and by modulating systemic levels of IL-6 and C-reactive protein (CRP) [15]. Similarly, a direct link between the gut and the lungs has been extensively documented. Dysbiosis is implicated in various pulmonary disorders, including COPD, cystic fibrosis, and asthma [8,16]. In a recent murine study, Lai et al. identified a causal relationship between gut microbiota alterations and COPD development using a cigarette smoke-induced COPD mouse model and fecal microbiota transplantation (FMT). The severity of COPD symptoms was significantly modulated by gut microbiota composition, and a commensal bacterial strain (*Parabacteroides goldsteinii*) was able to attenuate inflammation and lung pathology [17]. Clinical evidence further supports this relationship: analysis of stool samples from a cohort of 99 COPD patients and 73 healthy controls revealed that the COPD group exhibited a reduced relative abundance of Bacteroidetes and an increased proportion of Firmicutes compared to healthy individuals [18]. Moreover, murine models underlined that inhaled agents, in particular cigarette smoke, may directly alter gut microbiota composition and impair intestinal barrier function [19]. Consistently, early-life exposure to antibiotics, which disrupts gut microbial balance, has been associated with an increased risk of developing atopic diseases, including asthma [16]. Similarly, malnutrition and age-related dysbiosis in the elderly can lead to increased intestinal permeability and systemic inflammation, contributing to the onset or worsening of chronic respiratory diseases [20].

In addition to dietary and microbiota-derived factors, inhaled environmental pollutants (PM_2.5_ and PM_10_) have the potential to directly impact the liver. Epidemiological evidence demonstrates that elevated PM_2.5_ levels are associated with significant increases in serum liver enzymes (alanine aminotransferase, aspartate aminotransferase, gamma glutamyl transferase), suggesting hepatocellular toxicity [21]. Specific constituents of particulate matter, such as nitrates and black carbon (BC), act as carriers of toxic compounds. Once inhaled, ultrafine particles can cross the alveolar–capillary barrier and enter systemic circulation [22]. These particles subsequently accumulate in the liver, where they interact with hepatocytes and Kupffer cells [23]. Mechanistically, particle translocation induces excessive reactive oxygen species (ROS) formation, activates pro-inflammatory pathways, and disrupts lipid metabolism, thereby promoting steatosis, fibrosis, and progression to cirrhosis or HCC [24]. Interestingly, the lung-liver axis is a bidirectional interplay. In severe inflammatory conditions, such as acute respiratory distress syndrome, liver dysfunction impairs the modulation of systemic and pulmonary inflammation: its role in endotoxin clearance, cytokine and eicosanoid regulation, and acute-phase protein synthesis becomes a critical determinant of respiratory outcomes [25]. Further, in infectious contexts such as pneumonia, mitochondrial dysfunction in lung venular capillaries triggers neutrophil-mediated liver congestion, illustrating how pulmonary injury can directly compromise hepatic perfusion via the lung-liver axis [26].

Figure 1 provides a graphical synthesis of the main molecular mechanisms connecting the lung–gut–liver axis and how air pollution can affect it. Further details are available in the next sections.

## 3. Impact of Air Pollutants on the Lung and Gut Microbiome

### 3.1. Impact of Air Pollution on Respiratory Tract Microbiome

Growing evidence indicates that exposure to air pollutants disrupts both the lung and gut microbiota, contributing to local and systemic inflammation via the gut-lung axis. The lungs, as the primary site of exposure, are directly affected by pollutants that alter the pulmonary microbiota and trigger local inflammatory responses. In a cohort of healthy young adults, Zhao et al. demonstrated that elevated levels of ambient PM_2.5_ were significantly associated with changes in the nasopharyngeal microbiota, with an increase in phyla such as Acidobacteria and Gemmatimonadetes, as well as the genus *Symbiobacterium* (phylum Firmicutes). These taxa are not typical of the upper respiratory tract and potentially reflect environmental impact [27]. Similarly, Mariani et al. found that short-term exposure (3 days) to ambient PM (PM_10_ and PM_2.5_) in healthy adults was linked to a notable increase in the nasal genus *Moraxella*. This was the only taxon showing a positive correlation with PM concentrations, whereas commensal genera such as *Corynebacterium* and *Staphylococcus* decreased significantly [2]. A recent systematic review by Arca-Lafuente et al. including 486 individuals from 9 different studies confirmed that chronic exposure to airborne PM, particularly PM_2.5_ and PM_10_, is associated with a significant disruption of the upper airway microbiota in adults. Specifically, a decrease in commensal and potentially protective taxa such as *Corynebacterium* and *Dolosigranulum* (both within the phylum Actinobacteria) was observed, together with an enrichment of genera including *Prevotella*, *Leptotrichia* and *Fusobacterium*, which are commonly associated with inflammatory and pathogenic profiles [28]. Moreover, pollution-induced microbiota disorders may exacerbate some pre-existing pulmonary diseases. Thus, exposure to NO_2_ has been associated with structural changes in the oropharyngeal microbiota of adults with asthma, including the enrichment of *Rothia*, *Actinomyces*, *Fusobacterium*, and *Leptotrichia* [29].

### 3.2. Impact of Air Pollution on Gut Microbiome

Beyond the lungs, inhalation of urban coarse particles has been shown to induce not only pulmonary inflammation but also colonic oxidative stress, highlighting the bidirectional nature of the lung–gut axis [30]. Preclinical studies on murine models of chronic inhalation of air pollutants have been instrumental in elucidating the mechanistic impacts of air pollution on both the lung and gut microbiomes. Notably, Mutlu et al. demonstrated how PM_2.5_ and NOx can significantly alter the composition and functionality of small bowel and colon microbiota. These particles can reach the gastrointestinal tract from the upper airway via mucociliary clearance and swallowing, directly impacting the gut ecosystem [31]. Subchronic inhalation of PM_2.5_ may induce significant alterations in gut microbial diversity and community structure, including reductions in beneficial genera such as *Lactobacillus* and *Bifidobacterium* and increases in potentially pathogenic taxa like Proteobacteria [32]. Similarly, after 10 weeks of subchronic inhalation of UFPM (~300–350 μg/m^3^; 6 h/day, 3 days/week), exposed mice exhibited significant shifts in intestinal microbial composition, including increased abundance of genera such as *Ruminococcaceae* UCG-014, *Alistipes*, and *Lachnospiraceae* NK4A136, alongside reduced levels of *Bacteroides* and *Muribaculum* [33]. Also, an in vivo study on adult mice exposed to PM_2.5_ generated from brake pad wear via intratracheal instillation revealed significant impairments in respiratory function and metabolic parameters, accompanied by systemic inflammation and early pulmonary fibrosis. Gut microbiota profiling showed marked dysbiosis, including enrichment of pro-inflammatory genera such as *Odoribacter* and *Tuzzerella*, and depletion of beneficial taxa like *Desulfovibrio* and *Butyricimonas* [34].

Although PM is the most studied, a wide range of other ambient pollutants significantly contribute to gut microbiome alterations and host responses. Cheng et al. conducted a multiomic investigation in a London-based cohort examining short-term exposure to traffic-related air pollution, which includes a wide range of pollutants such as PM, NOx, Carbon Monoxide, O_3_. They observed significant shifts in the relative abundance of key bacterial taxa, including a decrease in *Faecalibacterium prausnitzii* and *Roseburia* spp., both of which are known butyrate producers, alongside increased levels of *Akkermansia muciniphila* and *Bacteroides* species. Metabolomic profiling has further revealed disruptions in SCFA biosynthesis and bile acids [35]. Elevated organophosphorus pesticides exposure has been linked to altered abundance of key bacterial taxa, including reductions in *Faecalibacterium* and *Bacteroides*, and increases in *Escherichia*/*Shigella* and *Ruminococcus gnavus* [36]. Similarly, exposure to polychlorinated biphenyls (especially PCB52), which are high carcinogenic chemical compounds formerly used in industrial applications, induces significant compositional shifts characterized by decreased abundance of beneficial taxa such as *Lactobacillus* and *Bacteroides*, and increased levels of inflammation-associated genera including *Oscillospira* and *Ruminococcus* [37].

### 3.3. Effects of Air Pollution-Induced Dysbiosis on Health and Disease

Alterations in the gut microbiota induced by air pollution may result in compromised intestinal barrier function and modified metabolomic profiles, potentially exerting adverse effects on systemic metabolism. Thus, wild-type mice exposed to PM_10_ exhibited elevated pro-inflammatory cytokine secretion and enhanced gut permeability [38]. In a murine model, a multi-omics analysis showed that gut microbiota alterations induced by chronic exposure to environmental PM_2.5_, such as the depletion of *Oscillospira* and *Helicobacter*, were accompanied by significant increases in gut-derived metabolites including L tryptophan, serotonin, and spermidine. These metabolic changes were paralleled by shifts in the lung metabolome. In particular, they involved purine metabolism (e.g., guanosine, hypoxanthine) and oxylipin signaling (e.g., hepoxilin B_3_), which are known to promote oxidative stress and inflammatory signaling. Collectively, these alterations compromised epithelial barrier integrity and amplified local and systemic inflammatory responses [39].

Furthermore, air pollution has been implicated in disrupting the intestinal redox lipidome, thereby affecting the production of free fatty acids and lipid metabolites essential for maintaining gut homeostasis. Atmospheric ultrafine particles have been associated with significant alterations in lipid metabolites in the mouse small intestine. Specifically, this exposure modulates the levels of both saturated and polyunsaturated fatty acids, while concurrently increasing lipid peroxidation products. Such mediators include bioactive eicosanoids derived from arachidonic acid metabolism which are key mediators of inflammation and oxidative stress [40]. Li et al. confirmed that microbiota imbalance due to UFPM exposure (i.e., a reduction in beneficial taxa such as *Lactobacillus* and *Bifidobacterium* and an enrichment in pro-inflammatory bacteria including Proteobacteria) leads to increased intestinal permeability. Interestingly, they observed that such shifts were associated with elevated blood levels of atherogenic lipid metabolites [41]. Moreover, PM_2.5_ inhalation causes an impaired glucose metabolism by disrupting SCFA production. In a murine model, Shao et al. showed that antibiotic-mediated depletion of the gut microbiota prevented the development of the metabolic phenotype associated with PM_2.5_ exposure, which was characterized by impaired glucose metabolism. Conversely, FMT from PM_2.5_-exposed donors was sufficient to reproduce this phenotype in germ-free mice [42]. Moreover, by promoting oxidative stress and inflammation, PM_2.5_ exposure impairs hepatic insulin signaling. Both in vitro and in vivo, Lu et al. observed increased reactive oxygen species, decreased antioxidant enzymes like Superoxide Dismutase 1 and Sirtuin 1, and higher levels of inflammatory cytokines such as IL-6 and tumor necrosis factor alpha (TNF-α), which together interfered with the insulin pathway and led to insulin resistance [43]. Collectively, these changes can worsen liver injury, enhance systemic inflammation, and disrupt lipid handling, ultimately resulting in a significant increase in atherogenic risk [44].

In addition, maternal exposure to PM_2.5_ has been shown to induce oxidative stress, inflammation, and disruption of colonic tight junctions in offspring. Liu et al. observed that such negative effects were alleviated by quercetin supplementation through modulation of the gut microbiota, particularly by reducing *Bacteroides* abundance [45]. Also, Bailey et al. investigated the impact of postnatal exposure to ambient air pollutants on the infant gut microbiota at 6 months of age in a longitudinal birth cohort. Their findings revealed that higher levels of PM_2.5_ and NO_2_ were significantly associated with alterations in the infant gut microbial community structure, including decreased abundance of beneficial genera such as *Bifidobacterium* and *Lactobacillus*, and increased levels of *Enterobacteriaceae* [46]. In the Spanish Maternal Microbes birth cohort including 162 babies, Cruells et al. investigated the role of environmental exposure during the first year of life. They found that prenatal and early postnatal exposure to NO_2_, BC, PM_2.5_ and O_3_ reduced α-diversity, and elevated NO_2_ acute and chronic exposure was associated with increased relative abundances of *Haemophilus*, *Akkermansia*, *Alistipes*, *Eggerthella*, and *Tyzerella* in infant feces [47].

## 4. Oxidative Stress as a Central Mechanism in Pollution-Induced Liver Damage

Oxidative stress serves as a pivotal mediator of liver injury resulting from exposure to PM_2.5_. Current evidence indicates that PM_2.5_ not only modifies the gut–liver axis but also directly disrupts hepatic metabolism, redox homeostasis, and immune signaling [48]. In this section we focus on the main molecular mechanisms underlying the relationship between air pollutants-related oxidative stress and liver disease, which mainly come from preclinical studies. Clinical data are summarized in the next section, which delves into liver disease in human beings.

A key initiating event is intestinal dysbiosis induced by chronic PM_2.5_ exposure. Zhao et al. demonstrated in mice that inhalation of PM_2.5_ results in a marked reduction in beneficial bacteria such as *Lactobacillus* and *Bifidobacterium*, and an expansion of pro-inflammatory genera including *Desulfovibrio* and *Oscillibacter*. This microbial imbalance leads to increased gut permeability and facilitates lipopolysaccharide translocation into the portal circulation, with upregulation of key inflammatory mediators including IL-1β, IL-18 and caspase. The subsequent activation of NOD-like Receptor Pyrin Domain-Containing 3 (NLRP3) inflammasome signaling in the liver ultimately drives inflammation and fibrosis [3]. Supporting these findings, Xiao et al. demonstrated that exposure to PM_2.5_ in both murine and cellular models not only induces hepatic steatosis and fibrosis, but also triggers the activation of hepatic stellate cells and lipid dysregulation as a consequence of gut-derived inflammatory stimuli [49].

Furthermore, PM_2.5_ exerts direct hepatotoxic effects, primarily by increasing oxidative stress and impairing lipid metabolism. Inhalation experiments conducted by Liu et al. revealed that exposure to PM_2.5_ impairs endoplasmic reticulum stress responses and disrupts lipid and glucose metabolism [50]. In human plasma samples from the UK Biobank, Aimuzi et al. identified a proteomic profile associated with chronic pollutant exposure, marked by the upregulation of immune mediators such as Tumor Necrosis Factor Receptor Superfamily Member 1B, Chemokine (C-C motif) Ligand 3 (CCL3), and C-X-C Motif Chemokine Ligand 9 (CXCL9), all of which are implicated in hepatic inflammation and fibrosis [51]. In another in vivo study, Xu et al. found that subchronic PM_2.5_ exposure (113 μg/m^3^, 12 weeks) induced hepatic oxidative stress and lipid metabolism dysfunction, hallmarks of MASLD. These effects were associated with increased malondialdehyde levels, reduced antioxidant enzyme activity, and dysregulation of key genes involved in lipid metabolism, including upregulation of Sterol Regulatory Element-Binding Protein 1c (SREBP-1c) and Fatty Acid Synthase (FAS), and downregulation of Peroxisome Proliferator-Activated Receptor Alpha (PPARα) [52]. These findings were supported by metabolomic data from a study by Yang et al., which showed the depletion of key antioxidants (Reduced Glutathione, Nicotinamide Adenine Dinucleotide Phosphate) and disruption of redox-sensitive pathways including glutathione turnover, purine metabolism, and the tricarboxylic acid cycle [53]. Indeed, mitochondrial dysfunction adds another layer of vulnerability, considering the major role of these organelles in ROS production and hepatic-specific anabolic pathways such as de novo lipogenesis and gluconeogenesis. Interestingly, long-term exposure to PM_2.5_ in rats was shown to induce ultrastructural mitochondrial damage, along with altered expression of circular RNAs linked with host genes coding for proteins involved in ubiquitination, zinc-ion binding, peroxisome function, and mitochondrial regulation [26].

PM_2.5_ also disrupts bile acid balance in the liver resulting in intrahepatic bile acid accumulation, which promotes oxidative stress. Zhang et al. showed that in hepatocyte cultures, some components of PM_2.5_ such as benzo[a]pyrene and cadmium bind to and activate the farnesoid X receptor (FXR), a nuclear receptor involved in bile acid regulation. This activation reduces the expression of cholesterol 7 alpha-hydroxylase (CYP7A1) and increases the levels of bile acid transporters ABCB11 and ABCC2, thereby altering both bile acid synthesis and transport [54]. These effects were confirmed in vivo by Yan et al., who exposed female C57BL/6 mice to ambient PM_2.5_ and found a clear disruption in hepatic metabolism. Metabolomic data showed an accumulation of lipids and primary bile acids, while transcriptomic analysis revealed increased expression of genes involved in bile acid and steroid synthesis, including Cyp7a1, Cyp27a1, small heterodimer partner (Shp), and fibroblast growth factor receptor 4 (Fgfr4) [55]. The resultant intrahepatic build-up of bile acids and lipids is intrinsically hepatotoxic, as it triggers oxidative and endoplasmic reticulum stress and activates inflammatory and fibrogenic signaling pathways, thereby promoting hepatocellular injury and fibrosis [54,55].

Multiple converging molecular pathways are involved in liver injury deriving from oxidative stress. ROS generated by PM_2.5_ initiate lipid peroxidation, leading to the formation of reactive aldehydes such as malondialdehyde (MDA) and 4-hydroxynonenal (4-HNE). These molecules form adducts with proteins and DNA, amplify mitochondrial dysfunction, and stimulate hepatic stellate cell activation, which is central to fibrogenesis [56]. Finally, considering these oxidative and metabolic injury pathways, recent studies have highlighted how chronic exposure to PM_2.5_ may ultimately contribute to hepatocarcinogenesis through a combination of genotoxic damage and epigenetic alterations. Indeed, PM_2.5_ can carry harmful substances like polycyclic aromatic hydrocarbons and heavy metals, which form DNA adducts, cause double-strand DNA breaks, and interfere with DNA repair mechanisms such as recombinase methylation—events that are known to promote carcinogenesis [57]. In addition, exposure to ambient PM has been associated with abnormal DNA methylation patterns [58].

The main mechanisms of PM_2.5_ hepatotoxicity are summarized in Table 1.

## 5. Air Pollution and Liver Disease: From Steatosis to Cancer

### 5.1. Air Pollution and MASLD

MASLD is the latest term used to describe steatotic liver disease associated with metabolic syndrome, replacing the previous term of non-alcoholic fatty liver disease (NAFLD). The new nomenclature has been mainly introduced to stress the strong link with pathogenetic risk factors such as diabetes, obesity and dyslipidemia, extending beyond the exclusion of alcohol as a main contributor [59]. MASLD encompasses a broad spectrum of hepatic injuries ranging from hepatic steatosis, metabolic dysfunction-associated steatohepatitis (MASH) and liver fibrosis, eventually resulting in cirrhosis and HCC.

Being the most common cause of chronic liver disease and liver-related morbidity and mortality, with a growing prevalence in recent decades, there is an urgent need to elucidate its pathogenesis, which is recognized to be complex and multifactorial. As we summarized in a recent review about the role of nicotine in MASLD [4], the “two-hit pathogenetic model” described MASLD as the result of two events. The first event is responsible for de novo lipogenesis and subsequent steatosis, the second for oxidative stress and lipid peroxidation. This theory, proposed by Berson et al. in 1998 [60], already highlighted the role of oxidative stress in liver disease. Currently, the “two-hit model” has been replaced by a “multiple hits” hypothesis, which encompasses multiple factors acting simultaneously, but still emphasizes the oxidative stress. Hepatic lipid accumulation generally triggers oxidative stress, reduces hepatic adenosine triphosphate (ATP) production and activates pro-inflammatory cytokines. These events may cause necroinflammation and drive progression to MASH [61].

Interestingly, a Mexican study conducted by Ruiz-Lara K. et al. observed that chronic exposure to ambient air and water pollutants was significantly associated with elevated levels of lipoperoxidation, generating oxidative stress and affecting the health status of people living in polluted areas [62]. Indeed, air pollution is a significant public health issue and an important risk factor for metabolic syndrome and MASLD, but the specific underlying mechanisms are still poorly understood in human beings, with limited data coming from preclinical studies. For example, a study conducted in Iran by Sepehri et al. on rats exposed to ambient air pollution (AAP) showed that airborne exposure to PM leads to the production of ROS and initiates inflammatory responses by inducing oxidative stress and causing deleterious health effects on animals. The liver catalase, malondialdehyde and glutathione peroxidase activities of the AAP rats were higher than those of the control group, reflecting increased oxidative stress [63]. Also, the authors found that air pollution induces histopathological liver alterations and the expression of apoptosis-related genes. The apoptosis effect seems to be triggered by O_3_ through increased expressions of genes such as caspase 8 (CASP8), CASP9, CASP3, and B-cell lymphoma 2 (BCL2), while SO_2_ induces the expression of B CL2-associated X protein (BAX), BCL2, CASP3, CASP8, and CASP9 apoptotic pathway in the liver tissue [63].

Evidence about human beings confirms the detrimental role of air pollution on liver disease. In a large-scale cross-sectional study conducted in Taiwan among more than 131.000 individuals undergoing health checkups, a non-linear association was observed between the exposure to six common air pollutants and the presence of MASLD. 40,6% of involved subjects were found to be affected by MASLD, defined as the combination of hepatic steatosis and at least one out of five cardiovascular criteria (overweight, diabetes/altered fasting serum glucose, blood hypertension, hypertriglyceridemia, hypercholesterolemia). PM_2.5_, PM_10_, O_3_, CO, and NO_2_ were positively associated with MASLD within specific concentrations ranges [64]. Guo et al. conducted an epidemiological cross-sectional study on more than 90.000 Chinese participants, concluding that long-term exposure to PM_1_, PM_2.5_, PM_10_, and NO_2_ was associated with an increased prevalence of NAFLD. Factors such as male sex, alcohol intake, cigarette smoking, high-fat diet and central obesity increased the negative effect of air pollution [65]. However, these studies, although including a considerable number of participants, do not allow any conclusions about causal relationships due to their cross-sectional design.

Interestingly, similar results derive from two large cohort studies, one Taiwanese cohort study by Chen et al. including more than 62,000 individuals undergoing health examinations between 1996 and 2016 [66], as well as a wide population-based cohort analyzed by Kong et al. involving more than 417,000 subjects from the UK biobank, which were recruited between 2006 and 2010 and followed-up overtime [67]. Chen et al. confirmed that people exposed to PM_2.5_, CO and NO_2_ had a higher HR for NAFLD, diagnosed by ultrasound [66]. Kong et al. analyzed the impact of air pollutants together with multiple lifestyle factors, which contributed to creating a composite lifestyle score. They identified unhealthy lifestyle as the main risk factor for NAFLD, with an additive negative effect due to the interaction between lifestyle and air pollution [67]. A very recent systematic review and meta-analysis by He et al., including seven cohort and seven cross-sectional studies, confirmed that exposure to NOx, PM_2.5_, PM_10_, PM_1_, PM_2.5–10_, passive smoking, and pollutants from solid fuels can increase the risk of NAFLD and its related cirrhosis, while exposure to O_3_ can reduce the risk. In developing countries, the risk of NAFLD and hepatic cirrhosis due to PM_2.5_ exposure is higher than in developed countries [5]. Started before the introduction of the new term MASLD, these studies use as their main outcome the previous definition of NAFLD, which did not require the presence of metabolic comorbidities. These findings highlight the need for further investigations to gather new specific data about MASLD and better clarify the complex relationship between air pollution and liver injury leading to MASLD, as well as the distinct pathogenic roles that different air pollutants may play in disease development.

### 5.2. Air Pollution and HCC

HCC is a major global health concern, showing a constantly increasing incidence [68]. Environmental factors play a significant role in its pathogenesis, and air pollution has been identified as a potential contributor. However, while it is clearly linked to several other adverse health outcomes, the potential association between air pollution and liver cancer remains unclear. Currently, many studies are examining the potential association between air pollution and primary liver cancer, but it is still difficult to draw causal conclusions. For example, Sun et al. did not find any association between air pollution and HCC in either European or East Asian populations, although a causal relationship was identified between NOx and a biomarker of hepatocellular differentiation, Arginase 1 [69]. In a recent systematic review, Gan et al. have checked the most relevant literature on the topic to identify possible associations between air pollution and liver cancer. They included thirteen cohort studies, globally involving almost 11,000,000 subjects. The most frequently examined component of air pollution was PM_2.5_, followed by NO_2_ and NOx, while fewer studies investigated other pollutants, such as PM_2.5_ absorbance, PM_10_, PM_2.5–10_, O_3_ and BC. A strong association was reported between PM_2.5_ and liver cancer mortality, but not liver cancer incidence. Surprisingly, no positive associations were detected regarding other air pollutants [70].

The main studies on main studies linking air pollution and MASLD/HCC are listed in Table 2.

### 5.3. Differences in Susceptibility to Air Pollution

Certain populations may be more susceptible to the development of MASLD and, more generally, to liver injury related to air pollution exposure. Although literature on this topic is still limited, some studies suggest that specific groups—including elderly people, children, and specific ethnicities—may be particularly vulnerable to the harmful effects of inhaled microparticles.

Thus, progression to liver fibrosis seems also to be increased among people affected by obesity, who have nearly double odds of fibrosis [71]. In a recent observational study, Schenker et al. investigated the associations between ambient air pollution exposure (PM_2.5_, PM_10_, NO_2_, O_3_) and hepatic fat fraction (HFF) and liver stiffness in Latino youth with obesity. The Latino population has a higher prevalence of MASLD than other racial/ethnic groups, which can be attributed to the high prevalence (approximately 50%) of a C-to-G polymorphism (rs738409) in the patatin-like phospholipase domain-containing protein 3 (PNPLA3) within this population [72]. Short-term O_3_ exposure was positively associated with HFF, whereas other pollutants showed no significant effects. These associations were not modified by PNPLA3 genotype or liver disease severity, indicating that O_3_ exposure may contribute to hepatic steatosis independently. Long-term exposure to 8hrMax O_3_ and redox-weighted oxidative capacity (Oxwt) was associated with increased liver stiffness, with stronger effects in adolescents carrying the PNPLA3 GG risk genotype than in those carrying the CC/CG genotypes. Additionally, youth with MASLD and established fibrosis demonstrated increased liver stiffness in response to short-term NO_2_ exposure, indicating that fibrotic livers may be more susceptible to pollutant-induced oxidative stress induced by environmental pollutants. These findings highlight ambient air pollution as a modifiable risk factor for hepatic steatosis and fibrosis in obese Latino youth. The results show interaction between genetic susceptibility (PNPLA3 GG genotype), disease severity, and environmental exposure, emphasizing the need for targeted interventions and research into pathways linking air pollutants to liver injury in vulnerable pediatric populations.

Evidence concerning older adults is particularly scarce; however, recent data suggest increased vulnerability in this population. In a community-based cohort of over 23,000 individuals aged ≥ 65 years with MASLD in China, 15.5% were classified as at high risk for liver fibrosis [73]. Exposure to elevated levels of O_3_, CO, SO_2_, PM_2.5_, and PM_10_ was significantly associated with an increased likelihood of fibrosis, whereas no association was observed for NO_2_. Analysis of combined pollutant exposures indicated that the overall risk was primarily driven by PM_10_. Factors such as sex, body mass index, diabetes, and physical activity modulated these associations, with inactive, obese, smoker and male individuals demonstrating greater vulnerability. These findings support the hypothesis that elderly patients with MASLD are particularly susceptible to the fibrogenic effects of air pollution, with PM_10_ emerging as the main determinant of risk.

## 6. Future Perspectives

Considering the substantial evidence of air pollution-induced dysbiosis and inflammation within the gut–lung axis, a growing number of studies indicate that microbiota modulation and antioxidant supplementation may offer protective benefits against pollutant-mediated tissue injury and systemic immune disturbances. Luo et al. demonstrated that supplementation with *Lactobacillus acidophilus* significantly attenuates PM-induced intestinal and pulmonary inflammation by restoring gut microbial balance, enhancing tight junction expression, and reducing systemic levels of pro-inflammatory cytokines such as IL-6 and TNF-α [74]. Recently, in a randomized, double-blind, placebo-controlled clinical trial, Wu et al. investigated the effects of a 12-week probiotic intervention on individuals chronically exposed to PM_2.5_. Participants received a daily dose of 10^10^ colony-forming units of a multi-strain probiotic formulation containing *Lactobacillus* and *Bifidobacterium* species. The intervention significantly improved lung function, measured by FEV_1_ and Forced Vital Capacity (FVC), alongside enhanced quality of life scores compared to placebo. Gut microbiota analysis via 16S rRNA sequencing revealed an increased relative abundance of anti-inflammatory genera such as *Lactobacillus*, *Bifidobacterium*, and *Faecalibacterium*, coupled with a reduction in pro-inflammatory taxa including *Enterobacteriaceae*. These microbiota shifts correlated with decreased systemic inflammatory markers (e.g., CRP and IL-6), supporting a mechanistic link through the gut-lung axis [75]. The targeted supplementation with prebiotics such as fructo-oligosaccharides and galacto-oligosaccharides and probiotics including strains of *Lactobacillus rhamnosus*, *Bifidobacterium longum*, and *Lactobacillus acidophilus*, can restore gut microbial balance disrupted by air pollution exposure in children. Symbiotic formulations combining these prebiotic fibers with probiotic strains have demonstrated synergistic effects, improving intestinal epithelial barrier function by enhancing tight junction protein expression (e.g., occludin and claudin-1), reducing gut permeability, and modulating systemic inflammatory responses [76]. Relatively new probiotics, the so-called “next-generation probiotics”, notably *Akkermansia muciniphila* and *Faecalibacterium prausnitzii*, are currently raising particular interest due to their potential therapeutic applications. The special attention towards these bacteria is attributable to data supporting their decreased abundances in the gut microbiome in NAFLD/MASLD, as we described above.

In addition to probiotics, other microbiota-modulating strategies have been attempted to mitigate air pollution effects on liver health by interfering with lung–gut–liver axis. For example, evidence on animal models shows that the use of antibiotics may be protective against diet-induced weight gain and adipocyte expansion, by upregulating the expression of genes involved in fat oxidation and thermogenesis [77]. The role of FMT is controversial, with some studies describing reduced fat accumulation and serum lipid levels after FMT, especially in non-obese patients [78]. However, many issues are still open, such as safety concerns, especially in immunocompromised patients or donor selection process [79]. Although promising and fascinating, such data about the therapeutic effect of microbiota modulation on air pollution-induced liver injuries are still preliminary and far from clinical practice, encumbered by numerous uncertainties that require further research for clarification.

## 7. Limitations

Although substantial progress has been made in elucidating the role of the lung–gut–liver axis in pollution-related liver disease, our current knowledge is still encumbered by important limitations. Firstly, most of the available mechanistic evidence derives from preclinical animal or cellular models. These systems provide invaluable insights but do not fully reproduce the complexity of human exposure scenarios, such as lifelong pollutant accumulation, genetic heterogeneity, and the variability of the human microbiome. Translational gaps therefore limit the extrapolation of preclinical findings to human pathology. Moreover, the available epidemiological studies are often constrained by indirect exposure assessment methods, relying on regional air quality monitoring rather than individual-level exposure. Such approaches may lead to exposure misclassification and underestimate the contribution of confounding variables, including lifestyle, diet, socioeconomic factors, and comorbid conditions. Regarding antioxidant and microbiota-modulating interventions, encouraging results observed in experimental models have not yet been translated into consistent clinical benefits. Available clinical trials are scarce, frequently underpowered, and heterogeneous in design. Unresolved questions include the optimal choice of compounds, dosing regimens, treatment duration, and long-term safety. Finally, reliable and non-invasive markers of oxidative stress, gut dysbiosis, or pollutant-related liver injury are lacking, hampering early detection and risk stratification. Novel technologies such as multi-omics profiling and artificial intelligence-based analysis may help to overcome these challenges, but validation in large, longitudinal human cohorts is still needed.

## 8. Conclusions

This review highlights the complex interactions along the lung–gut–liver axis, mediating the detrimental effects of air pollution on liver health. PM_2.5_, one of the most studied airborne pollutants, plays a central role in inducing oxidative stress, microbiota dysbiosis, metabolic imbalance, and chronic inflammation. These mechanisms converge to impair hepatic function and may contribute to the progression from MASLD to cirrhosis and HCC. Accumulating evidence supports the involvement of gut-derived inflammatory signals, mitochondrial dysfunction, and epigenetic modifications in pollution-driven liver injury. Emerging interventions such as antioxidant supplementation and microbiota-targeted therapies—including next-generation probiotics and symbiotics— may offer promising strategies to counteract pollution-induced liver damage. Further research is warranted to validate these approaches and to define personalized prevention strategies for populations at risk.

Beyond mechanistic insights, current evidence underscores the importance of translating these findings into clinical practice and public health strategies. For individuals chronically exposed to high levels of air pollution—particularly those with metabolic comorbidities such as obesity or diabetes—combined surveillance with liver function tests and non-invasive imaging could allow earlier detection of MASLD and fibrosis. Risk prediction models for MASLD could be strengthened by explicitly including air pollution exposure, particularly PM_2.5_, as an additional variable together with traditional metabolic risk factors such as obesity, diabetes, and dyslipidemia. Moreover, at the population level, policy measures including urban greening, stricter air quality standards, and air purification interventions should be evaluated through cost–benefit analyses that explicitly consider liver disease outcomes, in addition to respiratory and cardiovascular health. Finally, special attention should be directed toward vulnerable groups such as children, elderly, and immune-compromised patients, who are more susceptible to pollution-related injury and may benefit froIn a large-scale cross-sectional study conducted in Taiwan among more than 131.000 individuals undergoing health checkups, a non-linear association was observed between the exposure to six common m targeted preventive programs. Incorporating these recommendations into clinical and policy frameworks may help bridge the gap between experimental findings and real-world prevention.

## Figures and Tables

**Figure 1 antioxidants-14-01148-f001:**
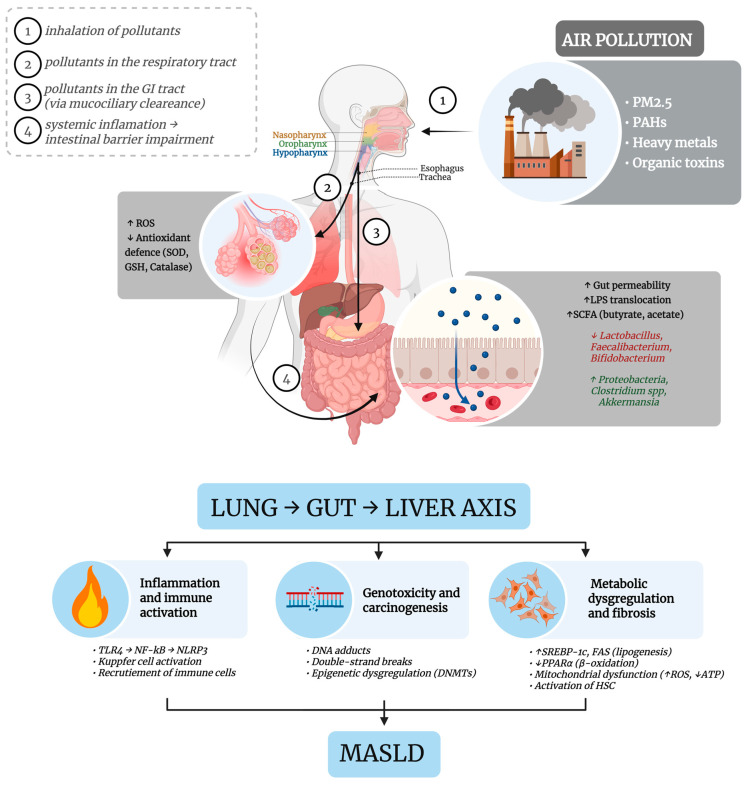
Schematic representation of air pollution-induced liver injury via Lung–Gut–Liver Axis: Microbiota, Metabolites, and Inflammatory Pathways. Inhaled fine particles generate ROS in the lungs and are translocated to the gastrointestinal tract, where they alter microbiota composition, increase intestinal permeability, and reduce SCFA production. These changes result in systemic translocation of microbial metabolites and PM_2.5_-associated toxins to the liver, where they activate inflammatory and fibrogenic pathways, alter lipid metabolism, and promote mitochondrial dysfunction and genotoxic stress—contributing to MASLD progression and potential carcinogenesis. Arrows indicate the direction of proposed mechanistic pathways and causal relationships between events. Abbreviations: ATP: Adenosine Triphosphate; DNA: Deoxyribonucleic Acid; DNMTs: DNA methyltransferase; FAS: Fatty Acid Synthase; GSH: Glutathione (Reduced Form); HSC: Hepatic Stellate Cell; LPS: lipopolysaccharide; NF-kB: Nuclear Factor kappa-light-chain-enhancer of activated B cells; NLRP3: NOD-like Receptor Pyrin Domain-Containing 3; PAHs: polycyclic aromatic hydrocarbons; PM_2.5_: particulate matter smaller than 2.5 µm; PPARα—Peroxisome Proliferator-Activated Receptor Alpha; ROS: Reactive Oxygen Species; SCFA: Short-Chain Fatty Acids; SOD: Superoxide Dismutase; SREBP-1c: Sterol Regulatory Element-Binding Protein; TLR4: Toll-Like Receptor 4.

**Table 1 antioxidants-14-01148-t001:** Summary of key mechanisms of PM_2.5_ hepatotoxicity in animal models.

Scheme	Mechanism of Hepatotoxicity	Animal Model	Key Findings
Fu et al., 2025 [3]	Gut–Liver axis impairment	C57BL/6 mice	Dysbiosis, ↑LPS, NLRP3 activation
Xiao et al., 2024 [49]	Gut–Liver axis impairment	C57BL/6 mice + cells	Gut-derived inflammation, HSC activation
Yang. et al., 2025 [50]	Oxidative Stress & altered lipid metabolism	BALB/c mice	IRE1α S-nitrosylation, ER dysfunction
Xu et al., 2019 [52]	Oxidative stress & altered lipid metabolism	ICR mice	↑ROS, ↓SOD/CAT, lipid dysregulation
Liu Y. et al., 2024 [53]	Oxidative stress & altered lipid metabolism	C57BL/6 mice	↓GSH/NADPH, disrupted TCA cycle
Liu Y. et al., 2024 [26]	Mitochondrial disfunction	Sprague–Dawley rats	Mitochondrial damage, circRNA changes
Zhang D. et al., 2024 [54]	Bile Acid Metabolism	Mice hepatocytes	FXR activation, ↓CYP7A1, ↑ABCB11/ABCC2
Yan et al., 2024 [55]	Bile Acid Metabolism	C57BL/6 female mice	↑Bile acids, ↑Cyp7a1, Shp, Fgfr4
Gan et al., 2024 [57]	Carcinogenesis (Nrf2 overactivation)	Human HCC samples, in vitro	↑Immune evasion, ↑chemoresistance via NQO1/HO-1

↑ indicates an increase; ↓ indicates a decrease. Abbreviations: ABCB11, ATP-binding cassette subfamily B member 11; ABCC2, ATP-binding cassette subfamily C member 2; AKT, protein kinase B; BALB/c, Bagg Albino laboratory-bred strain c; CAT, catalase; circRNA, circular RNA; C57BL/6, C57 Black 6 mouse strain; Cyp7a1, cholesterol 7 alpha-hydroxylase; CYP7A1, cholesterol 7 alpha-hydroxylase; ER, endoplasmic reticulum; Fgfr4, fibroblast growth factor receptor 4; FXR, farnesoid X receptor; GSH, reduced glutathione; HCC, hepatocellular carcinoma; HepG2, human hepatocellular carcinoma cell line; HO-1, heme oxygenase 1; HSC, hepatic stellate cell; ICR, Institute of Cancer Research mouse strain; IL-6, interleukin 6; IRE1α, inositol-requiring enzyme 1 alpha; LPS, lipopolysaccharide; NADPH, nicotinamide adenine dinucleotide phosphate (reduced form); NLRP3, NOD-like receptor pyrin domain-containing 3; NQO1, NAD(P)H quinone dehydrogenase 1; Nrf2, nuclear factor erythroid 2–related factor 2; PI3K, phosphoinositide 3-kinase; ROS, reactive oxygen species; Shp, small heterodimer partner; SOD, superoxide dismutase; TCA cycle, tricarboxylic acid cycle; TNF-α, tumor necrosis factor alpha.

**Table 2 antioxidants-14-01148-t002:** Overview of key human studies exploring the link between air pollution and MASLD/HCC.

Reference	Study Design	Subjects (n)	Aim	Results
Cheng et al., 2024 [64]	Cross-sectional	131,592	To investigate the non-linear relationship between ambient air pollution and MASLD prevalence	Non-linear associations were observed between MASLD and 6 air pollutants
Guo et al., 2022 [65]	Cross-sectional	90.086	To investigate the association between long-term air pollution exposure and MASLD	Increased exposure levels to all 4 air pollutants (PM_1_, PM_2.5_, PM_10_, and NO_2_) were significantly associated with MASLD.
Chen et al., 2025 [66]	Cohort study	62.660	To investigate the association between long-term air pollution exposure and NAFLD	Increased risk of NAFLD in subjects exposed to PM_2.5_, NO_2_ and carbon monoxide
Kong et al., 2024 [67]	Cohort study	417.025	To assess the joint associations of air pollution and lifestyle with NAFLD	Lifestyle primary risk factor for NAFLD; significant additive interaction between air pollution and lifestyle
Paredes-Marin et al., 2025 [71]	Cross-sectional	463	To investigate the association between environmental factors and liver fibrosis	Air pollution exposure + obesity nearly double the odds of significant fibrosis
Ran et al., 2024 [21]	Observational and Mendelian randomization study	244.842	To identify metabolic signatures associated with air pollution exposure and to explore their associations with the risk of MASLD	PM_2.5_, PM_10_, NO_2_ and NOx-related metabolic signatures appear to be associated with MASLD.
Sun et al., 2023 [69]	Mendelian randomization study	471	To assess the causal relationship between air pollution and HCC	No statistical association between PM_2.5_, PM_10_, PM_2.5–10_, NO_2_ and HCC in European and East Asian populations, but causal relationship between NOx hepatocellular differentiation.

Abbreviations: MASLD, Metabolic Dysfunction-Associated Steatotic Liver Disease; NAFLD, non-alcoholic fatty liver disease; PM_2.5_, Particulate Matter ≤ 2.5 µm; PM_10_, Particulate Matter ≤ 10 µm; NO_2_, Nitrogen Dioxide; HCC, hepatocellular carcinoma.

## Data Availability

No new data were created or analyzed in this study. Data sharing is not applicable to this article.

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
