# Peer review of "The Impact of Air Pollution on the Lung–Gut–Liver Axis: Oxidative Stress and Its Role in Liver Disease"

_antioxidants, 2025, doi:10.3390/antiox14101148_

Round 1
Reviewer 1 Report
The manuscript by Iaccarino and coll. addresses the role of air pollution in oxidative stress and its implications for liver disease. The topic is of interest given the increasing attention to oxidative stress–related pathways in human health. The authors provide a broad overview of the literature, supported by a substantial reference list. However, in its present form, the manuscript is largely descriptive and lacks the critical analysis and originality that distinguish a high-quality review article.
Major concerns
- The manuscript summarizes existing studies but does not sufficiently evaluate their strengths, limitations, or methodological differences. I strongly recommend including a comparative and critical discussion rather than a simple narrative summary.
- Several reviews with overlapping content have already been published (for example: 10.1016/j.ecoenv.2024.117469; 1186/s12889-025-24058-4). The current manuscript does not clearly identify what unique perspective it offers beyond these. The authors should explicitly state in the introduction how their review differs from and complements previous works.
- Pathways are described in broad terms, but the mechanistic links between oxidative stress and liver disease are not explored in sufficient detail.
- Figures are schematic and generic. More mechanistically detailed and original illustrations would enhance the value of the review.
- The conclusion is brief and general. A strong “Limitations and Future Directions” section is missing. For example, what are the unresolved questions regarding the translation of antioxidant interventions into clinical benefit? Where do preclinical models fail to reflect human disease? How might novel biomarkers or technologies advance the field?
Minor comments
- The manuscript would benefit from professional English editing. Some sentences are grammatically incorrect or overly long, which reduces readability.
Author Response
The manuscript by Iaccarino and coll. addresses the role of air pollution in oxidative stress and its implications for liver disease. The topic is of interest given the increasing attention to oxidative stress–related pathways in human health. The authors provide a broad overview of the literature, supported by a substantial reference list. However, in its present form, the manuscript is largely descriptive and lacks the critical analysis and originality that distinguish a high-quality review article.
Major concerns
- The manuscript summarizes existing studies but does not sufficiently evaluate their strengths, limitations, or methodological differences. I strongly recommend including a comparative and critical discussion rather than a simple narrative summary.
R: according to Reviewer’s recommendation, we have revised the text to better clarify strengths and limitations of the included papers. We believe that the new version provides a more critical overview of the topic.
- Several reviews with overlapping content have already been published (for example: 10.1016/j.ecoenv.2024.117469; 1186/s12889-025-24058-4). The current manuscript does not clearly identify what unique perspective it offers beyond these. The authors should explicitly state in the introduction how their review differs from and complements previous works.
R: We thank the Reviewer for this comment pointing out the importance of clarifying the novelty of our review. We have revised the Introduction to explicitly highlight how our review differs from and complements the previously published works. See lin 56-60.
- Pathways are described in broad terms, but the mechanistic links between oxidative stress and liver disease are not explored in sufficient detail.
R: We thank the reviewer for this observation. In the revised manuscript, we have expanded Section 4 to provide a more detailed account of the molecular mechanisms linking oxidative stress to liver disease.
- Figures are schematic and generic. More mechanistically detailed and original illustrations would enhance the value of the review.
R: thank you for your suggestion. We have generated a new image to improve its quality.
- The conclusion is brief and general. A strong “Limitations and Future Directions” section is missing. For example, what are the unresolved questions regarding the translation of antioxidant interventions into clinical benefit? Where do preclinical models fail to reflect human disease? How might novel biomarkers or technologies advance the field?
R: We thank the Reviewer for this valuable suggestion. In line with the comment, we have added a dedicated section entitled “7. Limitations” which discusses the main challenges and unresolved questions. We believe this addition strengthens the manuscript by providing a more critical perspective and clarifying the main areas that warrant future investigation.
Minor comments
- The manuscript would benefit from professional English editing. Some sentences are grammatically incorrect or overly long, which reduces readability.
R: thank you. We thoroughly revised the manuscript to correct grammar mistake and improve text readability.
Reviewer 2 Report
This manuscript aims to review recent findings on the impact of air pollution on liver disease pathogenesis, exploring the molecular, genetic, and microbiome-related mechanisms underlying lung–gut–liver interactions, providing insights into potential strategies to prevent or mitigate liver disease progression. The topic is relevant to the journal’s scope, well-written, and supported by appropriate literature. I suggest minor points to improve the manuscript:
- I suggest to add some lines about the potential biases and lack of informations of the current literature about this topic;
- Add the Legend to explain the abbreviations and the use of arrows for Figure 1;
- Remove the use of italics for "spp.".
I suggest minor points to improve the manuscript:
- I suggest to add some lines about the potential biases and lack of informations of the current literature about this topic;
- Add the Legend to explain the abbreviations and the use of arrows for Figure 1;
- Remove the use of italics for "spp.".
Author Response
This manuscript aims to review recent findings on the impact of air pollution on liver disease pathogenesis, exploring the molecular, genetic, and microbiome-related mechanisms underlying lung–gut–liver interactions, providing insights into potential strategies to prevent or mitigate liver disease progression. The topic is relevant to the journal’s scope, well-written, and supported by appropriate literature. I suggest minor points to improve the manuscript:
- I suggest to add some lines about the potential biases and lack of informations of the current literature about this topic.
R: We agree with the Reviewer and, to address this point, we have expanded the manuscript by introducing a new section (7. Limitations), where we critically discuss the main methodological shortcomings of available studies and highlight the need for validated biomarkers and longitudinal human data.
- Add the Legend to explain the abbreviations and the use of arrows for Figure 1
R: thank you for the comment. We have added the abbreviations’ list to make the figure more easily accessible.
- Remove the use of italics for "spp.".
R: thank you. Italics removed.
Reviewer 3 Report
The authors reviewed recent studies about air pollution influence on liver disease, and highlighted that pollutants, such as PM2.5 and NOx, can be deposited in the airways and subsequently transported to the gut. This process leads to microbial imbalance and intestinal barrier function damage, which lead to the transport of gut-derived LPS and substances to liver via the portal vein. These transportation promote the progression of MASLD to liver fibrosis and HCC. The review also summarized the preliminary protective effects of probiotics, synbiotics, and antioxidant interventions in restoring microbial balance and alleviating liver injury. This is the first review to integrate oxidative stress and microbial imbalance with the lung-gut-liver axis. It provides new ideas for developing prevention and personalized treatment strategies for air pollution-related liver diseases based on microbiota modulation and antioxidant approaches. While many concerns should be addressed before the manuscript can be accepted for publication.
Major points:
- individuals carrying genetic polymorphisms associated with liver disease susceptibility (e.g., PNPLA3 and TM6SF2) may have less tolerant to air pollution. The article only sporadically mentions obese individuals with higher risk in air pollution-related liver disease, but does not systematically discuss children, pregnant women, individuals with metabolic syndrome, or those with genetic susceptibility (PNPLA3, TM6SF2 mutant) to air pollution-related liver disease. It is suggested that the authors summarize the differences in responses to air pollution among different genotypes and population subgroups.
- The review assumes that gut-derived products can reach the liver via the portal vein, but does not assess whether lung-derived particles/inflammatory mediators can also cross the blood-liver barrier through systemic circulation. It is recommended that the authors analyze whether lung-derived pollutants can directly affect the liver.
- The review primarily cites animal studies with limited clinical data in human. It is suggested to set a separate section for elaborating the impact of air pollutants on liver disease progression as found in clinical studies.
- The article lacks policy recommendations. It is suggested that the authors add practise recommendations and existing policy concerning air pollution high-risk populations at the end of the article, such as establishing combined liver function-imaging screening for air pollution high-risk populations; incorporating PM2.5 into MASLD risk scores; conducting cost-benefit assessments of urban greening/air purification interventions, to provide operational suggestions for public health or clinical practice.
Minor points:
- In Page 3-Line117,the author stated as <Figure 2 provides a graphical synthesis>, which may be considered as Figure 1, the only one figure in the manuscript.
- In Page 3-Line117,the author stated as <Figure 2 provides a graphical synthesis>, which may be considered as Figure 1, the only one figure in the manuscript.
Author Response
The authors reviewed recent studies about air pollution influence on liver disease, and highlighted that pollutants, such as PM2.5 and NOx, can be deposited in the airways and subsequently transported to the gut. This process leads to microbial imbalance and intestinal barrier function damage, which lead to the transport of gut-derived LPS and substances to liver via the portal vein. These transportations promote the progression of MASLD to liver fibrosis and HCC. The review also summarized the preliminary protective effects of probiotics, synbiotics, and antioxidant interventions in restoring microbial balance and alleviating liver injury. This is the first review to integrate oxidative stress and microbial imbalance with the lung-gut-liver axis. It provides new ideas for developing prevention and personalized treatment strategies for air pollution-related liver diseases based on microbiota modulation and antioxidant approaches. While many concerns should be addressed before the manuscript can be accepted for publication.
Major points:
- individuals carrying genetic polymorphisms associated with liver disease susceptibility (e.g., PNPLA3 and TM6SF2) may have less tolerant to air pollution. The article only sporadically mentions obese individuals with higher risk in air pollution-related liver disease, but does not systematically discuss children, pregnant women, individuals with metabolic syndrome, or those with genetic susceptibility (PNPLA3, TM6SF2 mutant) to air pollution-related liver disease. It is suggested that the authors summarize the differences in responses to air pollution among different genotypes and population subgroups.
R: We thank the reviewer for this interesting suggestion. As suggested, we have revised the manuscript by adding Section 5.3, where we discuss the influence of ethnic background, age, and clinical conditions (obesity, metabolic syndrome, pregnancy, and pediatric populations) on susceptibility to air pollution-related liver disease.
- The review assumes that gut-derived products can reach the liver via the portal vein, but does not assess whether lung-derived particles/inflammatory mediators can also cross the blood-liver barrier through systemic circulation. It is recommended that the authors analyze whether lung-derived pollutants can directly affect the liver.
R: we have implemented section 2 by introducing a specific paragraph focusing on direct impact on the liver by lung-derived mediators. Thus, the complex network connecting lung, liver and gut results more explicit.
- The review primarily cites animal studies with limited clinical data in human. It is suggested to set a separate section for elaborating the impact of air pollutants on liver disease progression as found in clinical studies.
R: we thank the reviewer for the comment. As we better clarified in section 4, current data about molecular mechanisms underlying the relationship between air pollutants and liver disease mainly come from pre-clinical studies. Clinical data on humans are summarized in section 5, so we believe that a separate section would be redundant.
- The article lacks policy recommendations. It is suggested that the authors add practise recommendations and existing policy concerning air pollution high-risk populations at the end of the article, such as establishing combined liver function-imaging screening for air pollution high-risk populations; incorporating PM2.5 into MASLD risk scores; conducting cost-benefit assessments of urban greening/air purification interventions, to provide operational suggestions for public health or clinical practice.
R: We thank the Reviewer for this important suggestion. In accordance with the comment, we have expanded the conclusions section by adding a dedicated paragraph on clinical and policy recommendation to provide a more operational perspective, aligning the review with practical implications for both clinicians and policymakers.
Minor points:
- In Page 3-Line117,the author stated as <Figure 2 provides a graphical synthesis>, which may be considered as Figure 1, the only one figure in the manuscript.
R: thank you, the number has been corrected.
Round 2
Reviewer 1 Report
The authors have satisfactorily addressed all major concerns. The revised manuscript is more critical, provides clearer novelty, includes detailed mechanistic insights, and is now suitable for publication.
The authors have adequately addressed the major concerns raised in the first round of review. The revised version of the manuscript now provides a more critical perspective on the literature, clearly outlines its novelty compared to previous reviews, and significantly expands the mechanistic discussion of oxidative stress pathways in liver disease. A new, more detailed figure has been added, and a dedicated Limitations and Future Directions section strengthens the critical appraisal of the field. Language and readability have also been improved.
Reviewer 3 Report
The authors have totally addressed my concern. The revised manuscript is suitable to publish.
The authors have totally addressed my concern. The revised manuscript is suitable to publish.